# Increased levels of immature and activated low density granulocytes and altered degradation of neutrophil extracellular traps in granulomatosis with polyangiitis

Spyridon Lipka[1,2], Lennard Ostendorf[2,3,4], Udo Schneider[4], Falk Hiepe[2,4], Falko Apel[1,5], Tobias Alexander[2,4] *

1 Department of Cellular Microbiology, Max Planck Institute for Infection Biology, Berlin, Germany,
2 Deutsches Rheuma-Forschungszentrum (DRFZ Berlin)–a Leibniz Institute, Autoimmunology Group, Berlin, Germany, 3 Department of Nephrology and Intensive Care Medicine–Charité–Universitätsmedizin Berlin, Freie Universität Berlin, Humboldt-Universität zu Berlin and the Berlin Institute of Health (BIH), Berlin, Germany, 4 Department of Rheumatology and Clinical Immunology–Charité–Universitätsmedizin Berlin, Freie Universität Berlin, Humboldt-Universität zu Berlin and the Berlin Institute of Health (BIH), Berlin, Germany, 5 Department of Biology, Humboldt University, Berlin, Germany

* tobias.alexander@charite.de

**Data Availability Statement:** All relevant data are within the manuscript and its Supporting Information files.

## Abstract

Granulomatosis with Polyangiitis (GPA) is a small vessel vasculitis typically associated with release of neutrophil extracellular traps (NETs) by activated neutrophils. In this study, we further aimed to investigate the contributions of neutrophils and NETs to the complex disease pathogenesis. We characterized the phenotype of neutrophils and their capacity to induce NETs. In addition, the level of circulating NETs, determined by neutrophil elastase/DNA complexes, and the capacity of patient sera to degrade NETs were investigated from blood samples of 12 GPA patients, 21 patients with systemic lupus erythematosus (SLE) and 21 healthy donors (HD). We found that GPA patients had significantly increased levels of low-density granulocytes (LDGs) compared to HD, which displayed an activated and more immature phenotype. While the propensity of normal-density granulocytes to release NETs and the levels of circulating NETs were not significantly different from HD, patient sera from GPA patients degraded NETs less effectively, which weakly correlated with markers of disease activity. In conclusion, increased levels of immature and activated LDGs and altered degradation of circulating NETs may contribute to pathogenesis of GPA, potentially by providing a source of autoantigens that trigger or further enhance autoimmune responses.

## Introduction

Granulomatosis with polyangiitis (GPA) is a pauci-immune vasculitis that affects small- and medium-size vessels [1]. It commonly affects the upper respiratory tract and in severe cases the lung and kidneys [2]. The immunopathogenesis of GPA is complex, involving cellular and soluble mediators of both the innate and the adaptive immune system [3]. The hallmark of the

**Funding:** This work was supported by core funding by the Max Planck Institute of Infection Biology, provided by the Max Planck Gesellschaft, and the Leibniz Science Campus Chronic Inflammation (www.chronischeentzuendung.org). The funders had no role in study design, data collection and analysis, decision to publish, or preparation of the manuscript.

**Competing interests:** No authors have competing interests

disease are antineutrophil cytoplasmatic antibodies (ANCAs), which are detectable in the vast majority of patients [4]. In GPA, ANCAs are mostly directed against the neutrophil protease proteinase 3 (PR3) [5], a serin protease with antimicrobial effects that is normally stored in granules of neutrophils [6]. These ANCAs have been shown to activate neutrophils to produce reactive oxygen species (ROS) and proteolytic enzymes, resulting in necrotizing vessel inflammation [7].

Recent studies have implicated neutrophils in the pathogenesis of GPA, particularly low-density granulocytes (LDGs) that displayed disease-specific characteristics [8] and transcriptional dysregulation [9,10]. Furthermore, LDGs in GPA and systemic lupus erythematosus (SLE) are more prone to spontaneously release NETs than neutrophils from healthy blood donors [11–14], and are regarded as a major source of NETs in ANCA-associated vasculitis (AAV) [15]. NETs result from a regulated necrotic death called NETosis, which is characterized by the extrusion of chromatin associated with antimicrobial molecules into the extracellular space [16]. They contain various proteins with proinflammatory characteristics, such as high-mobility group box 1 (HMGB1), neutrophil elastase (NE), and, notably, myeloperoxidase (MPO) and PR3 [17]. While NETs are commonly induced by pathogens during immediate host defence [18,19], immune complexes [20] and PR3- and MPO-ANCA are also capable of inducing NETosis [21].

It was previously shown that NETs were present in inflamed tissues, including glomeruli in kidney biopsies [21] and affected skin lesions [22] of AAV patients. In addition, NET-associated proteins and structures have been detected in the circulation of patients (reviewed in [15]), but the precise role of NETosis in GPA remains to be determined. It has been proposed that NETs contribute to vessel inflammation directly by damaging endothelial cells [12,23] and indirectly by triggering adaptive immune responses leading to generation of ANCAs [15]. NETs are degraded in the blood by an enzyme called DNase 1, an exonuclease found in serum [24]. Thus, overabundance of NETs in AAV may not only result from enhanced production but also from impaired clearance. Initial reports have already indicated a less efficient degradation of NETs in AAV due to impaired DNase 1 activity [12,25]. While the source of PR3 inducing the production of ANCA in GPA is not clearly identified, it is possible that PR3 contained within NETs [26] may play a role.

Herein, we further delineated the role of neutrophils and NETs in the complex immunopathogenesis of GPA. We found that levels of peripheral blood LDGs were significantly increased in GPA patients compared to healthy controls, and displayed an activated and more immature phenotype. The propensity of neutrophils to undergo NETosis was not enhanced when normal-density granulocytes (NDGs) were incubated *in vitro* and NETs, determined by DNA/NE complexes, were not significantly elevated in sera from GPA patients. However, the capacity of GPA sera to degrade NETs was significantly impaired in our cohort of patients and correlated with clinical and serologic markers of disease activity, implicating netting neutrophils as critical component for the disease pathogenesis of GPA.

## Methods

### Patients and controls

The study population included 12 patients with granulomatosis with polyangiitis (GPA) who fulfilled the 2017 Provisional ACR/EULAR classification criteria for GPA [27]. Their demographics disease characteristics of patients are provided in Table 1. 21 patients with systemic lupus erythematosus (SLE) meeting the 2019 EULAR/ACR classification criteria [28] and 21 healthy controls served as control. The study was approved by the Institutional Review Board

**Table 1. Patient characteristics.**

| | Healthy donors | GPA | SLE | *P* Value |
|---|---|---|---|---|
| Number | 21 | 12 | 21 | |
| Age (median, IQR) | 30 (20–56) | 58 (52–76) | 34 (18–52) | HD vs. GPA *P*<0.001 <br> HD vs. SLE *P* = 0.162 |
| Female (n, %): | 20 (95.0) | 5 (46.0) | 20 (95.0) | HD vs. GPA *P*<0.001 <br> HD vs. SLE *P* = 1.00 |
| Clinical manifestations (n, %) <br> Renal involvement <br> Head/Neck <br> Pulmonary | 0 <br> 0 <br> 0 | 8 (66.7) <br> 7 (58.3) <br> 8 (66.7) | 4 <br> 0 <br> 0 | |
| ANCA status <br> PR-3 positivity (n, %) <br> MPO positivity (n, %) | - <br> - | 9 (75.0) <br> 0 (0) | 0 <br> 0 | |
| BVAS (median score, range) | - | 6.5 (0–18) | - | |
| SLEDAI-2K (median score, range) | - | - | (2–24) | |
| Immunosuppressive and biologic medications (n, %) | 0 | GC: 11 (91.7) <br> AZA: 2 (16.7) <br> MTX: 5 (41.7) <br> RTX: 3 (25) | GC: 18 (85.7) <br> HCQ: 14 (66.7) <br> AZA: 5 (23.8) <br> MMF: 8 (38.1) <br> MTX: 2 (9.5) <br> CsA: 1 (4.8) <br> BEL: 1 (4.8) | |
| Daily prednisolone dosage (mg) (median, range) | 0 | 13.5 (0–50) | 10.0 (0–100) | |

Demographic and serological data of GPA and SLE patients. Abbreviations: AZA, Azathioprine; BEL, Belimumab; BVAS, Birmingham Vasculitis Activity Score; CsA, Ciclosporin A; GC, Glucocorticoids; HCQ, Hydroxychloroquine; MMF, Mycophenolate Mofetil; MPO, Myeloperoxidase; MTX, Methotrexate; PR3, proteinase 3; RTX, Rituximab; SLEDAI-2K, Systemic Lupus Erythematosus Disease Activity Index 2000. Statistical analysis comparing differences in age was performed using the Mann-Whitney test, sex differences with the chi-square test.

of the Charité –University Medicine Berlin (EA 1/372/13) and written informed consent was obtained by all participants prior to enrolment in the study.

## Cell preparation and separation

Neutrophils were freshly isolated from peripheral blood as previously described [29]. Briefly, heparinized peripheral blood was layered on Histopaque 1119 (Sigma-Aldrich) and centrifuged at 800g for 20min. After collecting and washing both the PBMC and neutrophil layer, the latter was subsequently layered on a discontinuous Percoll-gradient (Cytiva). After centrifugation at 800g for 20min, NDGs were harvested and washed before further analysis.

## Flow cytometry

To investigate the phenotype of neutrophils, freshly isolated NDGs and LDGs were incubated with the following antibodies for 30 minutes on ice: CD10-PerCP-Cy5.5 (clone HI10a, BD Pharmingen, 25μg/mL), CD15-PE-Vio770 (clone VIMC6), CD16-VioGreen (clone REA423), CD33-FITC (clone REA775), CD62L-VioBlue (clone 145/15), CD63-PE (clone REA1055) and CD66b-APC (clone REA306), all Miltenyi Biotec, Germany. Flow cytometry was performed using the MACSQuant® flow cytometer (Miltenyi Biotec, Germany). Data analysis was performed with FlowJo software (TreeStar, CA, version 10.6.1). To determine the absolute numbers of NDGs from peripheral blood, Truecount TM tubes (BD Biosciences) were used according to the manufacturer's instructions after staining full blood with CD45-FITC (clone REA747), CD16-VioGreen (clone REA423) and CD19-APC (clone REA675, all Miltenyi

Biotec) and lysis with immediate fixation using the Fix/Lyse Solution (Invitrogen). In contrast, absolute numbers of LDGs were determined by calculating the percentage of CD15[+] LDGs contained in the PBMC-layer from the absolute number of CD45[+] PBMCs as determined by Truecount assessment.

## NETosis assay

NET-production of peripheral blood neutrophils was determined *in vitro* as previously described [30]. Briefly, freshly isolated neutrophils were plated on coverslips and treated with 100 nM PMA (phorbol-12-myristate-13-acetate, Sigma Aldrich) in RPMI containing 0.05% human serum albumin (HSA, Gibco® ThermoFisher) or left untreated. After fixation with 2% PFA (paraformaldehyde) at 2 and 4h, the coverslips were washed and permeabilized with 0.5% TritonX-100/PBS for 5 min. Subsequently, cells were incubated for 2h with an anti-chromatin antibody (PL2-3, produced in house), a rabbit-anti-neutrophil-elastase-antibody (Calbiochem, Merck). After washing, goat anti-mouse Alexa568, goat anti-rabbit Alexa488 and Hoechst 3342 (all ThermoFisher) were added and incubated for 1 hour on ice. After several washing steps, immunofluorescence microscopy was performed using the Evos FL Auto 2 fluorescent microscope from Invitrogen. Images were analysed using ImageJ as described before. Briefly using the settings and thresholds described by Brinkmann et al. [30] the NET-rate was calculated as: NET-rate = 100*Objects counted (chromatin channel)/Objects counted (Hoechst channel).

## NET ELISA and Proteinase-3 ELISA

A previously developed ELISA, detecting DNA/NE complexes, was used to determine the amount of circulating NETs in peripheral blood [31]. Briefly, 50 µl cryopreserved EDTA-plasma from patients was added to anti-neutrophil elastase (NE)-precoated plates using the human anti-NE ELISA kit (Hycult Biotech, HK319-01). After incubation at room temperature for 2h on the shaker with agitation at 300 rpm, plates were washed with ELISA wash-buffer (PBS, 0.05% Tween-20). Subsequently, 50 µl of anti-DNA-POD-antibodies (Cell Death Detection ELISA[PLUS], Roche) were added and incubated for 1 hour. After washing and developing with ABTS and stopping with $H_2SO_4$, plates were measured in a microplate reader (VersaMax microplate reader[TM], Molecular Devices). A standard curve was created by a serial dilution using isolated NETs from HD previously stimulated with PMA. The level of NETs was determined using the NanoDrop 2000 device, measuring the amount of DNA at a wavelength of 280 nm. All analyses were performed in duplicate, and mean values were reported.

Anti-Proteinase 3 (PR3) antibodies were analyzed from freshly obtained serum samples in our central laboratory facility (Labor Berlin). Quantitative assessment of IgG antibodies directed against PR3 was performed by ELISA (ORG 618, Anti-PR3 hs, ORGENTEC Diagnostika, Germany).

## DNase 1 assay

The capacity of donor sera to degrade NETs was investigated as previously described [24]. Briefly, 0.4 µg NETs, induced with PMA in freshly isolated neutrophils from a healthy donor, were incubated with 50 µl of patient serum (25%) in the presence of 1U/ml Deoxyribonuclease 1 (DNase 1) or left untreated as negative control. The amount of DNA was determined at $t_0 =$ 0h and after incubation at $t_1 = 4{,}5$ h and overnight ($t_2 = 21$ h) with the Quant-iT[TM] Pico-Green[TM] dsDNA Assay-Kit (ThermoFisher). After calculating the percentages $P_1 = (t_1/t_0)*$ 100% and $P_2 = (t_2/t_0)*100\%$, the percentage of degraded NETs was defined at $t_1$ as $(100-P_1)$ % and at $t_2$ as $(100-P_2)$ %. The positive control (NETs treated with DNase 1) was set to be 100%

of degraded NETs and the data were normalized accordingly. All analyses were carried out in triplicate, and mean values were reported.

### Statistical analysis

Statistical analysis was conducted using GraphPad Prism 8.0.2 (San Diego, CA). For comparison of data between the patient groups, we used the Kruskal-Wallis test with Dunn's Correction for multiple testing. Correlation analysis of LDG frequencies, NET-production and NET-degradation with clinical and serologic disease markers was performed using Spearman correlation.

## Results

### LDGs are increased in GPA and display an immature and activated phenotype

We first investigated the number and phenotype of both low-density granulocytes (LDGs) and normal-density granulocytes (NDGs), freshly isolated from the peripheral blood of GPA patients, by flow cytometry. We found significantly increased levels of CD15$^+$SSC$^{hi}$ LDGs [8,32–34] in GPA patients compared to HD (median 7.7% vs 2.6%, $P$ = 0.020) (Fig 1A), while the absolute number of LDGs did not differ (Fig 1B). The gating strategy is shown for one representative HD (Fig 1C) and one GPA patient (Fig 1D). Likewise, frequencies and absolute numbers of circulating CD16$^+$ NDGs were not significant different between HD and GPA patients, when analyzed from whole blood samples (S1A Fig). To assess whether GPA neutrophils differed phenotypically from neutrophils of HD, we investigated their surface expression of markers associated with activation, differentiation, adhesion and degranulation. Compared to HD, LDGs from GPA patients showed a more immature phenotype, reflected by significantly lower expression levels of CD10 (MFI 2288 vs 4576, $P$ = 0.039, Fig 2A). In contrast, levels of CD33, another marker associated with immaturity, were not differentially expressed (Fig 2C). Furthermore, LDGs in GPA expressed significantly higher levels of CD66b (MFI 55605 vs 9724, $P$ = 0.006), indicating a more activated phenotype (Fig 2F). However, other markers associated with activation (CD16, Fig 2B), adhesion (CD62L, Fig 2D) and degranulation (CD63, Fig 2E) were not differentially expressed between GPA and HD, emphasizing the unique nature of these LDGs. Similarly, NDGs of GPA patients expressed lower amounts of CD10 (Fig 2A) and higher amounts of CD66b (Fig 2F), indicating a more general alteration of granulocyte phenotypes in GPA.

### NDGs in GPA display no enhanced NETosis *in vitro*

Neutrophils from patients with GPA have been reported to generate NETs more robustly than healthy individuals [32,33]. These studies most commonly investigated LDGs, which appear in the peripheral blood mononuclear cell layer of density-separated blood. To assess the capability of neutrophils to undergo NETosis in our cohort of patients, we purified NDGs from peripheral blood and measured their ability to produce NETs spontaneously and after stimulation with PMA. Representative immunofluorescence images displaying netting neutrophils are provided in S1B Fig. Without stimulation and after stimulation with PMA for 2 and 4 hours, respectively, no significant differences in the quantity of NET formation were observed (Fig 3), indicating that NDGs in GPA have no added propensity to produce NETs *in vitro* in our cohort of patients.

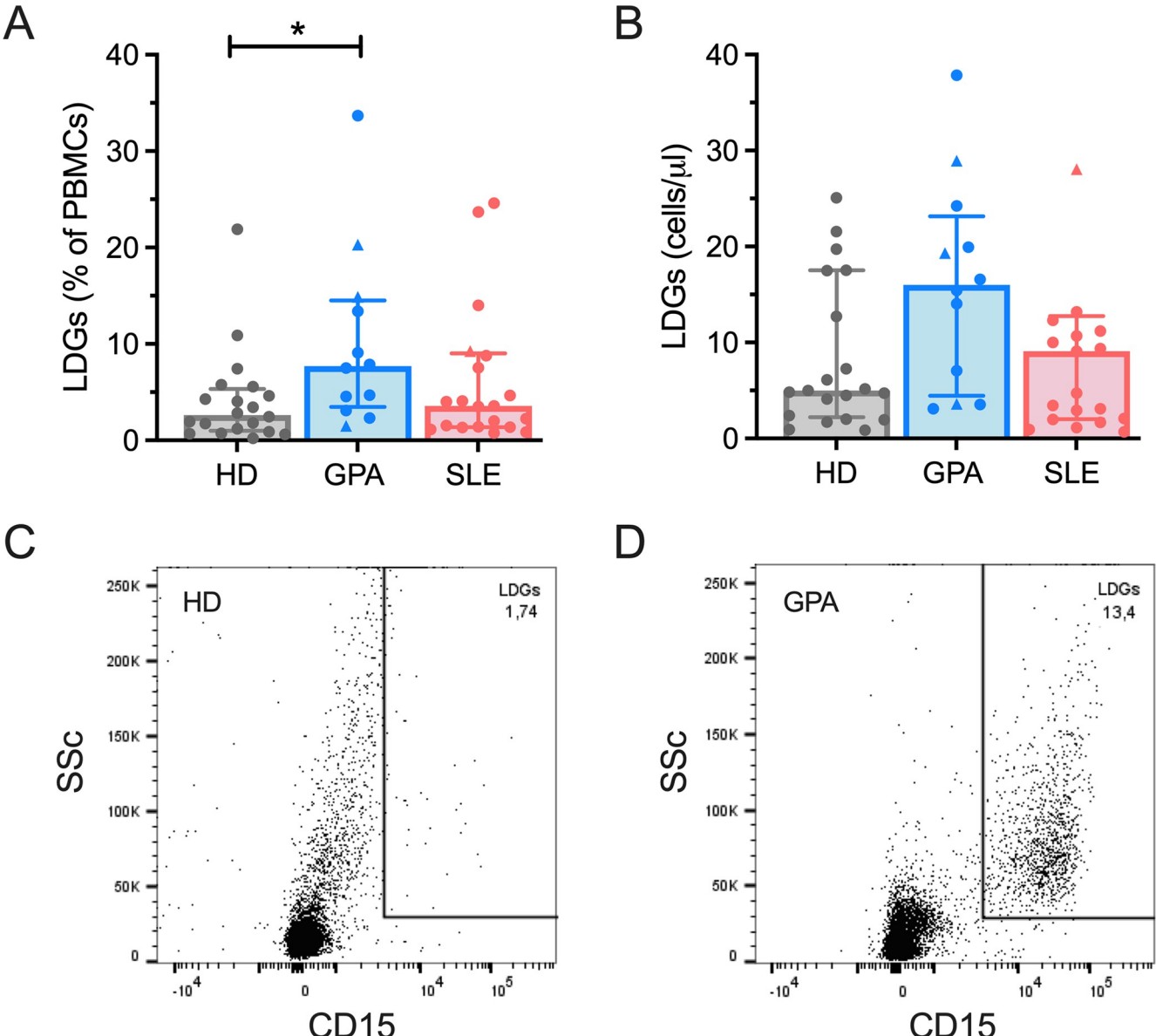

**Fig 1. Quantification of LDGs.** Percentage (A) and absolute numbers (B) of LDGs in peripheral blood (HD, n = 21; GPA, n = 12; SLE, n = 21). LDGs were defined as CD15+SSC$^{hi}$ cells in the PBMC-layer of Percoll-separated blood by FACS, as previously reported (33). Each dot represents one measured patient sample. Pateints receiving prednisolone dosages of $\geq$ 20mg daily (3 patients with GPA and one patient with SLE) are indicated by triangles. Bars indicate median, error bars represent interquartile range. Data were analyzed by Kruskal-Wallis-test ($\alpha$ = 0.05), *p<0.05, **p<0.01.Representative figures for the detetction of LDGs in a healthy donor (C) and a patient with GPA(D) are provided.

## Circulating NETs were not increased in GPA patients

Previous studies identified increased levels of NET remnants in the circulation of AAV patients, defined either as nucleosome/MPO complexes [21,35], total DNA or DNA/MPO complexes [36], or as mitochondrial DNA [37]. In this study, we measured the content of circulating NETs by using a previously developed ELISA detecting DNA/NE complexes, which was validated in a cohort of patients with *P. falciparum* malaria infection [31]. By using this assay, we found no increased levels of NETs isolated from EDTA-plasma of GPA patients compared to control groups (Fig 4A).

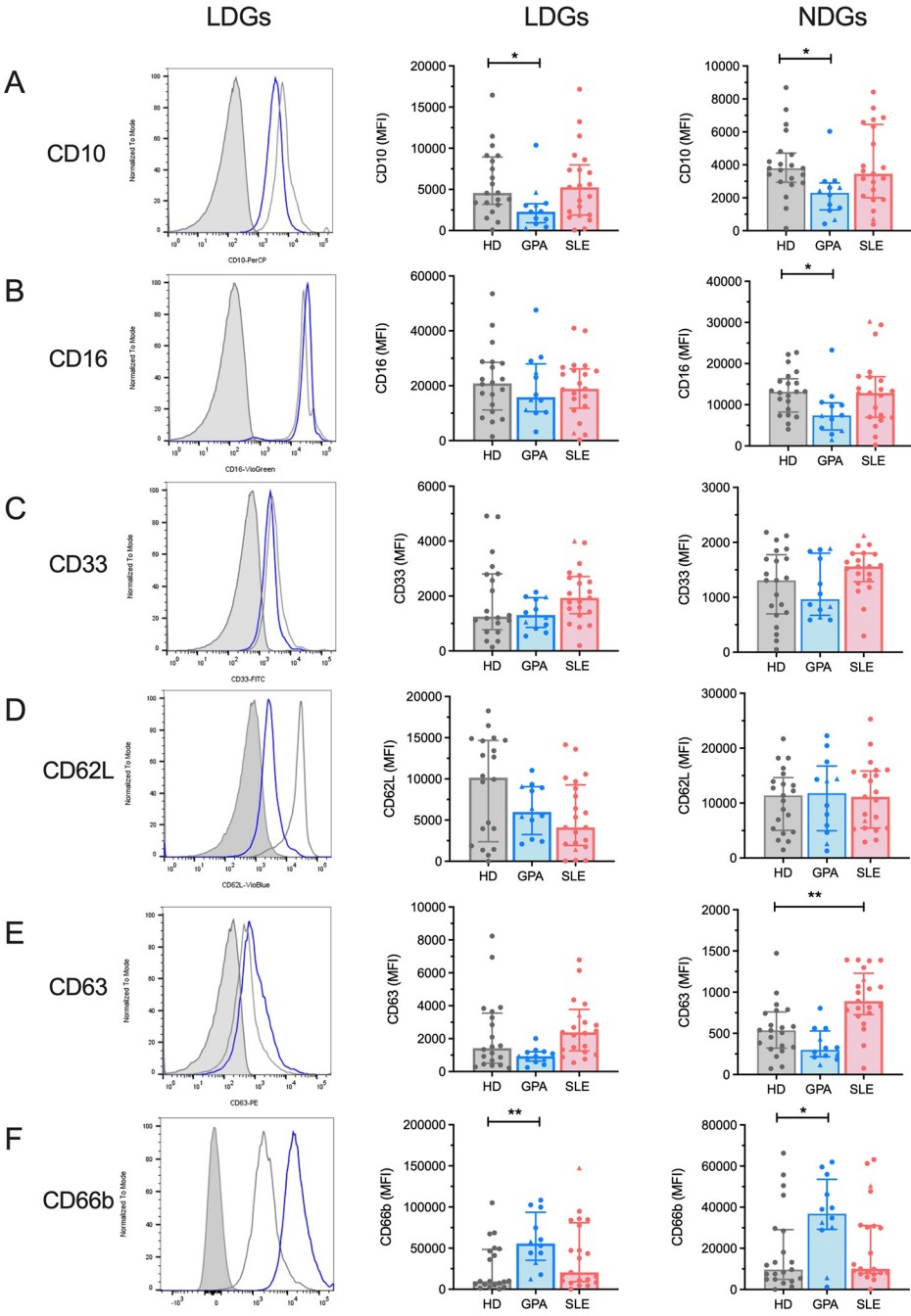

**Fig 2. Phenotypic analysis of LDGs and NDGs.** Surface markers associated with (A,C) maturity (CD10, CD33), (B,F) activation (CD16, CD66b), (D) adhesion (CD62L) and (E) degranulation (CD63) were measured on LDGs and NDGs (HD, n = 21; GPA, n = 12; SLE, n = 21) by flow cytometry. Pateints receiving prednisolone dosages of ≥ 20mg daily (3 patients with GPA and one patient with SLE) are indicated by triangles. Representative histograms of LDGs with respective FMO control stainings (grey filled curve) for a healthy donor (grey line) and a GPA patient (blue line) are shown. Each dot represents one measured patient sample. Median values ± IQR are presented. Data were analysed by the Kruskal-Wallis-test (α = 0.05), *p<0.05, **p<0.01.

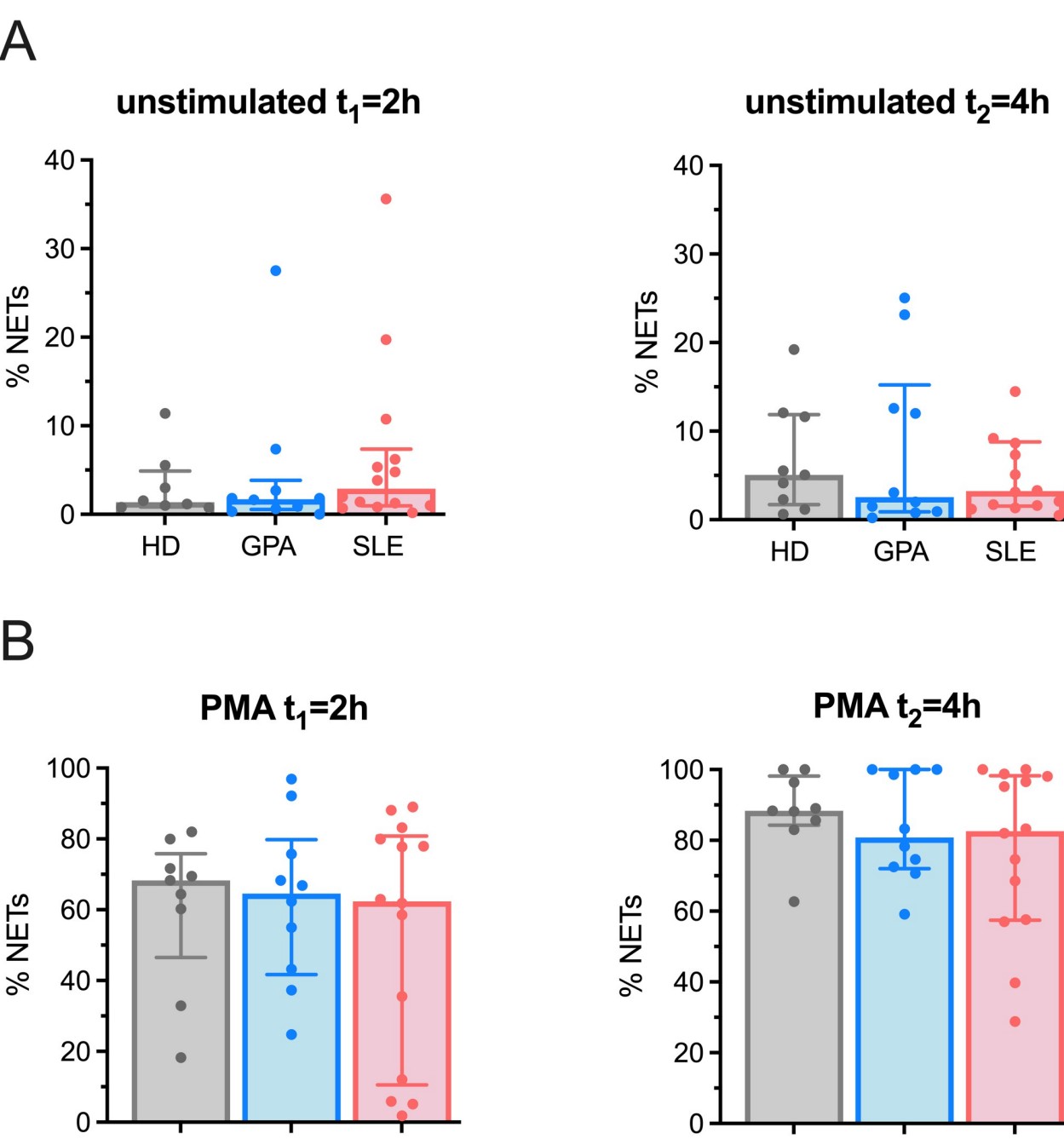

**Fig 3. Quantification of NET-production *in vitro*.** Freshly isolated NDGs were either (A) left untreated or (B) stimulated with 100nM PMA for 2 and 4 hours, respectively. The ability of NDGs to produce NETs was determined using immunofluorescence and a semiautomatic quantification in HD (n = 9), GPA (n = 10) and SLE (n = 14). Representative images are shown in S1B Fig and a detailed description of patient demographics is provided in S1 Table. Each dot represents one measured patient sample. Median values ± IQR are presented. Data were analysed by the Kruskal-Wallis-test (α = 0.05).

## Sera of GPA patients degrade NETs less efficiently than sera of HD

NET degradation involves digestion by macrophages and degradation by DNase 1, an endonuclease found in peripheral blood [24]. A previous study from our group demonstrated that a

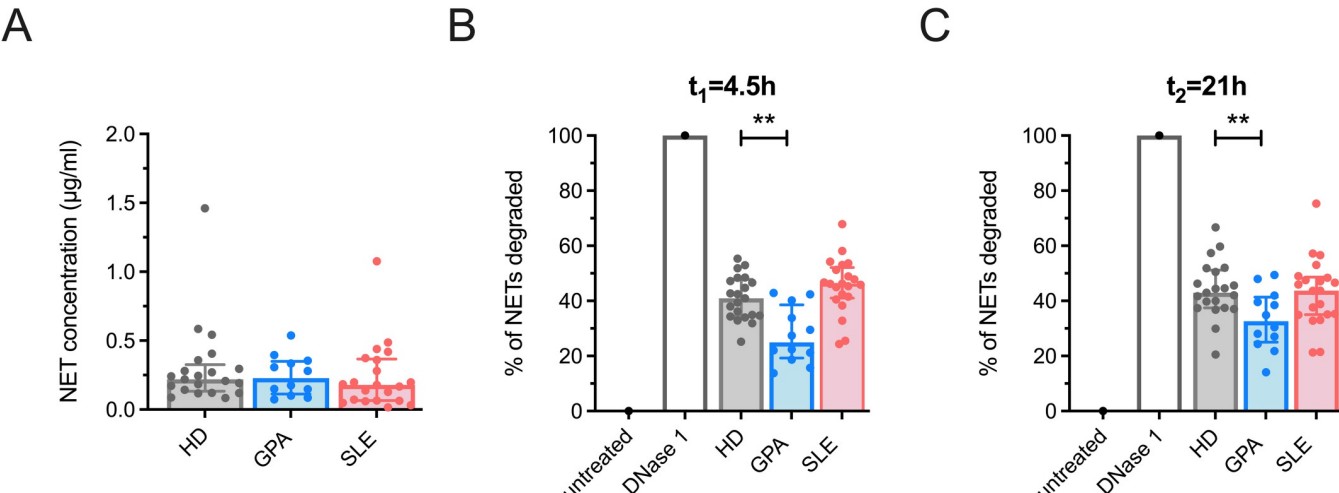

**Fig 4. Quantification of NETs isolated from peripheral blood and DNase 1 assay.** (A) NET-concentrations were determined by a NET-ELISA in EDTA-plasma (HD, n = 21; GPA, n = 12; SLE, n = 21). NET degradation activity of DNase 1 in serum was determined after $t_1 = 4,5$ h (B) and $t_2 = 21$ h (C) in HD, n = 21; GPA, n = 12; SLE: n = 21. Median values ± IQR are presented. Data were analysed by Kruskal-Wallis-test (α = 0.05), *p<0.05, **p<0.01.

subset of SLE patients degraded NETs poorly due to impaired serum DNase 1 function [24]. To determine the ability of patient sera to degrade NETs in GPA, we isolated NETs released by PMA-stimulated neutrophils from a healthy donor and incubated them with patient sera. The quenching of picogreen-fluorescence served as readout for NET degradation. We found that sera of the GPA-patients degraded NETs less efficiently than HD after incubation for 4.5 and 21 hours, respectively (Fig 4B and 4C), suggesting an underlying clearance deficiency in GPA.

## DNase 1 degrading ability correlates inversely with clinical parameters of GPA

To investigate the potential biologic implications of the detected neutrophil alterations in GPA, we correlated our findings with clinical and serologic features of the disease. We found that the impaired degradation of NETs weakly correlated inversely with the Birmingham Vasculitis Activity Score (BVAS), a clinical score that measures clinical activity of vasculitis (r = -0.582, $P$ = 0.051), and with serum levels of anti-PR3, which are associated with disease activity (r = -0.627, $P$ = 0.034) (Fig 5A). These findings might be confounded by glucocorticoid therapy, as there was a similar trend of reduced NET degradation in patients with higher prednisolone dosage. In contrast, neither NET-concentration in plasma (Fig 5B) nor capacity of NGDs to form NETs after PMA stimulation (Fig 5C) or frequency of peripheral blood LDGs (Fig 5D) significantly correlated with disease markers investigated.

## Discussion

There is a growing body of evidence that LDGs and NETs contribute to the pathogenesis of systemic autoimmune diseases, such as AAV [15,38]. In this study, we further investigated the role of neutrophils and NETs in the complex immune dysregulation of GPA and found that LDGs were significantly increased in peripheral blood and displayed a more immature and activated phenotype in GPA compared to healthy controls. Furthermore, while neither NET formation of NDGs nor circulating NETs were elevated, patient sera from GPA patients degraded NETs less effectively compared to HD. The impaired NET degradation inversely

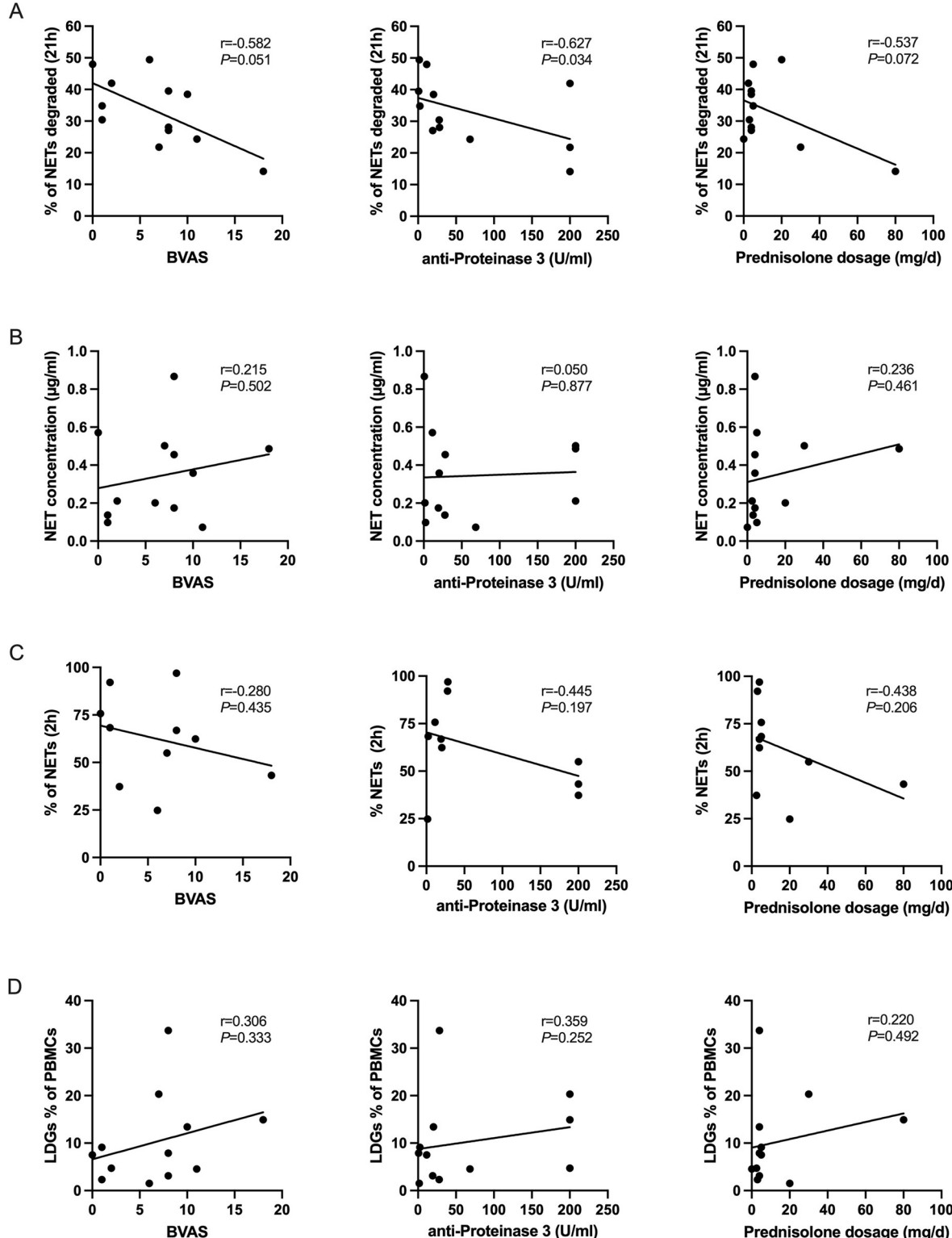

**Fig 5. Correlation of NET-degradation, NET-concentration, NET formation of NDGs, and periperal blood LDG levels with clinical and serologic activity measures in GPA, as well as daily prednisolone dosage.** Spearman-correlations of (A) NETs-degradation-activity after 21 hours of incubation, (B) NET-concentration in peripheral blood as measured by ELISA, (C) frequency of NET formation of NDGs after 2h of PMA stimulation and (D) LDG levels in peripheral blood with BVAS, anti-PR3 titer in serum, as well as daily prednisolone dosage. Each dot represents one measured patient sample. Correlation-coefficient and *P*-value are depicted in the figure (α = 0.05).

correlated with clinical and serologic activation markers of GPA, such as BVAS scores and serum ANCA levels, implicating a potential role in the immunopathogenesis of the disease.

Our data confirm and extend previous findings. Particularly, increased frequencies of circulating LDGs were previously reported by several groups investigating AAV [8,32] and linked to disease activity [8,39,40]. In line with these notions, our cohort of GPA patients was characterized by high clinical and serologic disease activity despite treatment with disease-modifying anti-rheumatic drugs (DMARDs), by means of elevated BVAS-scores and c-ANCA levels, and comprised a 3-fold higher fraction of LDG within PBMC compared to HD. Although the precise mechanisms contributing to the overabundance of LDGs in AAV remain elusive, recent data suggest that they are highly heterogeneous in phenotype and function, and predominantly display features of hyperactivity and immaturity, distinct from their NDG counterparts [8,39,40]. In accordance with these findings, LDGs in our GPA patients were characterized by significantly decreased expression levels of the maturity marker CD10 and increased levels of the activation marker CD66b, while displaying normal expression levels of markers associated with immaturity (CD33), activation (CD16), adhesion (CD62L) and degranulation (CD63), emphasizing the unique nature of LDGs in GPA. Notably, heterogeneous populations of both immature and activated neutrophils have also been described in patients with Graft versus Host disease [41], HIV infection [42], multiple sclerosis [33], and SLE [43], suggesting common immunologic pathways under systemic activation of neutrophils and/or an "emergency granulopoiesis", a process leading to increased release of neutrophils from the bone marrow during sepsis [44]. Similar mechanisms might therefore account for presence of such neutrophils in GPA.

Previous studies clearly linked neutrophils in GPA with more intense NETosis *in vitro*, particularly LDGs [11,12,15,39,40]. We now investigated the NET formation potential of NDGs, as they are by far the most abundant group of neutrophils in the circulation and because purification of LDGs with magnetic cell sorting (MACS), as described by other groups [43], has several limitations. Surprisingly, in contrast to previous reports, we found no increased propensity of NDGs to undergo NETosis, neither spontaneously nor upon stimulation. This discrepancy may be attributed to different methods of NET-detection, the influence of immunosuppressive therapies or the heterogeneity of clinical manifestations of included patients.

In addition to the presence of NETs in inflamed tissues from AAV patients [21,45], it has been demonstrated that these patients also have increased levels of NET-associated components in the circulation [46–54], both in active disease and remission [36]. In these studies, NETs were defined as nucleosome/MPO complexes [15,21], total DNA or DNA/MPO complexes [36], or as mitochondrial DNA [37]. We now measured the amount of circulating NETs by using a previously developed ELISA detecting DNA/NE complexes, that was previously used by our group to demonstrate increased NET release in patients with *P. falciparum* malaria infection [31]. By applying this method, we detected no significant differences in neutrophil-derived NETs in GPA-patients compared to HD and SLE. This was somewhat surprising in view of the fact that NE, in addition to MPO, has been demonstrated to regulate the formation of NETs [55], and previous studies measuring NE observed a correlation between NE and disease activity in AAV [56].

Increased NET formation must be balanced against clearance mechanisms, which involve macrophages, dendritic cells, and DNase 1 [57,58]. By incubating netting neutrophils derived from healthy donors with patient sera, we found that the degradation of NETs was significantly impaired in GPA patients compared to HD. These findings confirm previous data demonstrating a reduced DNA-degradation capacity in GPA [12], microscopic polyangiitis (MPA) [25] and SLE [12,24]. This effect could result from a reduced DNase 1 activity, as previously reported in MPA [25] and SLE patients [24,25]. However, whether this is linked to inherited

defects, like mutations [59] or polymorphisms [60] in DNase 1, predisposing to disease development, or the consequence of systemic inflammation and autoantibody formation remains unclear. Previous studies in SLE already indicated that antibodies binding to NETs significantly impaired DNase 1 degradation of NETs *in vitro* [24]. Reciprocally, studies in AAV suggested that ANCA may support the clearance of NETs, possibly through opsonisation and the formation of immune complexes [35].

Although GPA patient serum was deficient in NET degradation, NET levels were not elevated in GPA plasma. This could indicate that other pathways of NET degradation were activated that contribute to immunopathology, e.g. NET clearance by macrophages or mDCs. Macrophages for instance, have been described to degrade NETs in healthy donors [57]. In addition, this discrepancy may be explained by the accumulation of NETs in inflamed tissue lesions as described before in the kidney or even the lungs of patients [45,61]. Further research is required to confirm these notions.

Inhibition or degradation of NETs could resemble an interesting novel treatment target in AAV. For example, enhanced NETosis may be disrupted therapeutically with administration of DNase 1, which has been shown to be safe and tolerable in SLE patients [62] and discussed as potential treatment in AAV [63]. Other potential agents include the NE inhibitors Alvelestat and BAY 85–8501, as recently discussed as therapeutic option in MPO-AAV [63]. Future studies are warranted to evaluate the contributions of netting neutrophils to the pathogenesis of AAV and to identify novel therapeutic interventions from these disturbances.

Our study has few limitations. Most importantly, the sample size of GPA patients was rather low and the study probably underpowered to draw reliable conclusions. In addition, the patient cohort was heterogeneous in terms of background DMARD therapies and clinical phenotypes of the disease. This is of importance in view of a recent publication demonstrating that distinct disease patterns in GPA are associated with differences in NET formation and NET content, where LDGs from patients with head and neck manifestations had a particularly higher propensity to undergo NETosis [11]. A potential caveat for the interpretation of our data may also results from significant difference in age and sex distribution between GPA patients and control cohorts. In this context, previous studies indicated sex differences in neutrophil biology. For example, transcriptome profiling from healthy young adult females had a more activated/mature neutrophil profile characterized by enhanced type I interferon pathway activity, compared to neutrophils from male donors [64].

In conclusion, we identified increased levels of LDGs in active GPA patients, phenotypically characterized by immature and activated properties, with reduced surface expression of CD10 and increased expression of CD66b. NDGs from GPA patients were not prone to undergo NETosis more robustly compared to those from HD, neither spontaneously, nor upon PMA-stimulation, and NET-containing components, determined by DNA/NE complexes, were not elevated in plasma of GPA patients. However, a reduced capacity of NET degradation was evident in GPA, which negatively correlated with markers of clinical and serologic activity of the patients. Together, our data further support a concept, by which NETs may provide a source of autoantigens, due to prolonged exposure, for the generation of PR3-producing B- and plasma cells, potentially triggering or enhancing chronic autoimmune responses in GPA. Although the precise molecular mechanisms contributing to neutrophil activation and NET formation remain to be determined, NETs may resemble a novel future therapeutic target that merits further investigation.

## Supporting information

**S1 Fig. A) Quantification of NDGs.** (A) Frequencies of NDGs among PBMCs and (B) absolute numbers of NDGs were determined by flow cytometry (HD, n = 21; GPA, n = 12; SLE,

n = 21) using the Truecount method. NDGs were identified as $CD16^+SSC^+$ cells in whole blood. Each dot represents one measured patient sample. Patients receiving prednisolone dosages of $\geq$ 20mg daily (3 patients with GPA and one patient with SLE) are indicated by triangles. Median values ± IQR are presented. Data were analyzed by Kruskal-Wallis-test, no significant differences were determined. **B) Microscopy images of netting neutrophils.** Representative immunofluorescence images to identify NETs formed by normal-density granulocytes (NDG) from one healthy donor and one patient with GPA without stimulation (upper row) or after certain time-points after PMA stimulation.
(PDF)

**S1 Table. Patient characteristics in which NETs have been investigated as depicted in Fig 3.**
(PDF)

## Author Contributions

**Conceptualization:** Falk Hiepe, Falko Apel, Tobias Alexander.

**Formal analysis:** Spyridon Lipka.

**Funding acquisition:** Falko Apel.

**Investigation:** Spyridon Lipka, Lennard Ostendorf.

**Methodology:** Falk Hiepe, Falko Apel, Tobias Alexander.

**Project administration:** Spyridon Lipka.

**Resources:** Tobias Alexander.

**Supervision:** Lennard Ostendorf, Udo Schneider, Falk Hiepe, Falko Apel, Tobias Alexander.

**Validation:** Udo Schneider.

**Writing – original draft:** Spyridon Lipka, Tobias Alexander.

**Writing – review & editing:** Falk Hiepe, Falko Apel, Tobias Alexander.

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
