## [Decision Letter · Decision Letter 0]

29 Nov 2022

PONE-D-22-27387

Increased levels of immature and activated low density granulocytes and altered degradation of neutrophil extracellular traps in granulomatosis with polyangiitis

PLOS ONE

Dear Dr. Alexander,

Thank you for submitting your manuscript to PLOS ONE. After careful consideration, we feel that it has merit but does not fully meet PLOS ONE’s publication criteria as it currently stands. Therefore, we invite you to submit a revised version of the manuscript that addresses the points raised during the review process.

We consider that the manuscript lacks precision in various aspects, and the data presented need to be completed and strengthened. We invite you to submit a revised version of the manuscript that addresses all the points raised during the review process (see below Reviewer's comments).

We look forward to receiving your revised manuscript.

Kind regards,

Prof. Pierre Bobé

Academic Editor

PLOS ONE

Journal Requirements:

2. You indicated that you had ethical approval for your study. Please clarify whether minors (participants under the age of 18 years) were included in this study. If yes, in your Methods section, please ensure you have also stated whether you obtained consent from parents or guardians of the minors included in the study or whether the research ethics committee or IRB specifically waived the need for their consent.

 "This work was supported by core funding by the Max Planck Institute of Infection Biology, provided by the Max Planck Gesellschaft, and the Leibniz Science Campus Chronic Inflammation (www.chronischeentzuendung.org). " 

 "No authors have competing interests"

7. Your ethics statement should only appear in the Methods section of your manuscript. If your ethics statement is written in any section besides the Methods, please delete it from any other section. 

Reviewers' comments:

Reviewer's Responses to Questions

**Comments to the Author**

1. Is the manuscript technically sound, and do the data support the conclusions?

Reviewer #1: Partly

2. Has the statistical analysis been performed appropriately and rigorously? 

Reviewer #1: No

3. Have the authors made all data underlying the findings in their manuscript fully available?

Reviewer #1: Yes

4. Is the manuscript presented in an intelligible fashion and written in standard English?

Reviewer #1: Yes

5. Review Comments to the Author

Reviewer #1: This is a small inception cohort study examining NET production in GPA (n=12) and SLE (n=21) receiving a wide array of medications. The healthy donor control group was 95% female and much younger than the GPA cohort, making it difficult to separate out gender and age effects. There was significant variability in virtually all readouts, indicating that the study is underpowered to assess most parameters studied. The study confirms previous findings of LDG expansion and impaired NET degradation in GPA.

• The cohorts should be separated into those with active and remission disease (the BVAS range included 0 so remission patients were apparently included). It is likely that disease activity (and, more importantly, the high dose corticosteroids associated with this active disease, will have a big effect on NETosis).

• Note that, although table 1 describes the full cohort, studies examining NETs had fewer participants; these cohorts should be described in a supplementary table.

• Fig S1A: it’s not clear what the y-axis represents. Neutrophils in the PBMC layer are LDGs, but the figure refers to NDGs.

• I suggest that, for all figures, the individuals receiving high dose corticosteroids (say >20mg/day) are identified using, for example, open circles. Also, in addition to correlating NET degrading capacity with BVAS, figure 5 should also correlate against steroid dose in both SLE and GPA patients.

• Are flow cytometry data available for whole blood samples? This is important to clarify the effect of cell isolation on markers such as CD16 and CD63. Indeed, it is not clear why the results pertaining to LDGs are emphasised, given that NETOsis, the focus of the paper, was not studied in this cell population.

• The anti-PR3 antibody levels, rather than c-ANCA levels, should be reported. I presume that none were anti-MPO positive.

• Fig 4: There does appear to be a decline in NET degrading capacity with increasing disease activity, although BVAS is a poor means of quantifying the degree of vasculitic injury. Is the same observed with SLEDAI in the lupus participants? Again, it is important to consider the confounding effect of concurrent corticosteroid therapy, which may in fact be the main reason for this decline in NET degradation capacity.

• The authors comment on some unexpected null findings in the discussion, for example, no increased circulating NETs in GPA. The most likely reason for this is the fact that the study is underpowered, which should be alluded to in the discussion.

• The methods section contains descriptions of quantification of DCs, B cells and other cell types that have no counterpart in the results section. Were these experiments performed? If not, it is not necessary to describe the respective methods.

• The strongest result is the reduction in NET degradation capacity of GPA serum. Were DNAse-1 levels measured in serum? This would be an obvious next step in determining the mechanism of this observation.

6. PLOS authors have the option to publish the peer review history of their article (what does this mean?). If published, this will include your full peer review and any attached files.

Reviewer #1: **Yes: **Mark Little

---

## [Author Response · Author response to Decision Letter 0]

2 Feb 2023

Dear Editor,

enclosed please find the revised version of the above-mentioned manuscript. In a separate file (rebuttal letter) we respond to each point raised by the reviewer. 

According to editorial issues you raised we can confirm the following: 

1. Ethical: No minors (participants under the age of 18 years) were included in this study.

2. Financial disclosure: This work was supported by core funding by the Max Planck Institute of Infection Biology, provided by the Max Planck Gesellschaft, and the Leibniz Science Campus Chronic Inflammation (www.chronischeentzuendung.org). The funders had no role in study design, data collection and analysis, decision to publish, or preparation of the manuscript.

3. Competing interest: No authors have competing interests

4. Data Availability: The study's minimal data set are uploaded as Supporting Information files.

5. My ORCID iD (0000-0003-1193-0097) will be uploaded as requested.

We hope that revised manuscript has been modified to your satisfaction can be considered for publication. Looking forward to hearing from you at your earliest convenience, we remain with best regards.

Sincerely, 

Tobias Alexander, MD

Charité – University Medicine Berlin, Department of Rheumatology and Clinical Immunology, Charitéplatz 1, 10117 Berlin, Germany

---

## [Editor Report · Decision Letter 1]

27 Feb 2023

Increased levels of immature and activated low density granulocytes and altered degradation of neutrophil extracellular traps in granulomatosis with polyangiitis

PONE-D-22-27387R1

Dear Dr. Tobias,

We’re pleased to inform you that your manuscript has been judged scientifically suitable for publication and will be formally accepted for publication once it meets all outstanding technical requirements.

Kind regards,

Prof. Pierre Bobé

Academic Editor

PLOS ONE

---

## [Editor Report · Acceptance letter]

6 Mar 2023

PONE-D-22-27387R1 

Increased levels of immature and activated low density granulocytes and altered degradation of neutrophil extracellular traps in granulomatosis with polyangiitis 

Dear Dr. Alexander:

I'm pleased to inform you that your manuscript has been deemed suitable for publication in PLOS ONE. Congratulations! Your manuscript is now with our production department. 

Kind regards, 

on behalf of

Prof Pierre Bobé 

Academic Editor

PLOS ONE